# Exploring Molecular Signs of Sex in the Marine Diatom *Skeletonema marinoi*

**DOI:** 10.3390/genes10070494

**Published:** 2019-06-28

**Authors:** Maria Immacolata Ferrante, Laura Entrambasaguas, Mathias Johansson, Mats Töpel, Anke Kremp, Marina Montresor, Anna Godhe

**Affiliations:** 1Integrative Marine Ecology, Stazione Zoologica Anton Dohrn, Villa Comunale, 80121 Naples, Italy; 2Department of Marine Sciences, University of Gothenburg, Box 461, SE 405 30 Göteborg, Sweden; 3Clinical Genomics Göteborg, SciLifeLab, Medicinaregatan 1G, 413 90 Göteborg, Sweden; 4Gothenburg Global Biodiversity Centre, Box 461, SE 405 30 Göteborg, Sweden; 5Biological Oceanography, Leibniz Institute for Baltic Sea Research Warnemünde, Seestraße 15, 18119 Rostock, Germany

**Keywords:** diatom, sexual reproduction, *Skeletonema marinoi*, transcriptomics, meiosis, flagella, sex induced genes (SIG)

## Abstract

Sexual reproduction plays a fundamental role in diatom life cycles. It contributes to increasing genetic diversity through meiotic recombination and also represents the phase where large-sized cells are produced to counteract the cell size reduction process that characterizes these microalgae. With the aim to identify genes linked to the sexual phase of the centric planktonic diatom *Skeletonema marinoi*, we carried out an RNA-seq experiment comparing the expression level of transcripts in sexualized cells with that of large cells not competent for sex. A set of genes involved in meiosis were found upregulated. Despite the fact that flagellate gametes were observed in the sample, we did not detect the expression of genes involved in the synthesis of flagella that were upregulated during sexual reproduction in another centric diatom. A comparison with the set of genes changing during the first phases of sexual reproduction of the pennate diatom *Pseudo-nitzschia multistriata* revealed the existence of commonalities, including the strong upregulation of genes with an unknown function that we named Sex Induced Genes (SIG). Our results further broadened the panel of genes that can be used as a marker for sexual reproduction of diatoms, crucial for the interpretation of metatranscriptomic datasets.

## 1. Introduction

Sexual reproduction is a ubiquitous event in all eukaryotic lineages, although information on the molecular mechanisms at the base of this process are mostly derived from model organisms. Meiotic sex, with ploidy alternation, cell–cell recognition systems and fusion, originated in the ancestral eukaryotes, and it is widespread in unicellular organisms [1]. The increasing availability of sequenced genomes and the information on molecular mechanisms that regulate sexual reproduction in various organisms now allow for the reconstruction of the evolution of this fundamental process (e.g., References [2,3]). Sexual reproduction plays a fundamental role in diatom life cycles, unicellular microalgae that are important drivers of global biogeochemistry [4]. It is the phase in which genetic recombination occurs when haploid gametes are produced following meiosis, but it also represents the phase in which large-sized cells originate. A progressive cell size reduction during mitotic cell division is in fact a distinctive feature of diatom species [5]. This is due to the presence of a rigid siliceous cell wall (the frustule) made of two slightly unequal halves. Upon cell division, daughter cells always synthesize the inner half of the frustule, and this implies that the average cell size of the population progressively decreases. In many diatoms, large-sized cells are produced following sexual reproduction, in which the zygote (called “auxospore”), not surrounded by the rigid frustule, can expand and, within this stage, the large-sized initial cell is synthesized (Figure 1). Diatoms, which together with oomycetes and brown algae belong to the group of Stramenopiles, are broadly distinguished in two groups: the centrics, with radial symmetry, and the pennates, with bilateral symmetry [6]. A homothallic mating system (monoecious), in which morphologically and functionally different gametes are produced in a clonal culture, is reported for centric diatoms, while a heterothallic mating system (dioecious), in which sexual reproduction can only occur between cells of opposite mating type, is common in pennate diatoms [7]. In both centric and pennate diatoms, sex can be induced only when cells reach a species-specific size threshold (sexualisation size threshold (SST)), i.e., the cohorts of cells derived from large initial cells have to undergo a certain number of mitotic divisions to reduce their size before they become competent for sex (Figure 1).

The sexual phase in centric diatoms seems to be induced by external cues [5,7]. This is also the case for the centric planktonic diatom *Skeletonema marinoi*, where gametangia (cells that will produce gametes), motile gametes, and auxospores were obtained by transferring cultures to higher salinity conditions [8]. The pedigree fingerprinting of microsatellite markers showed that both inter- and intra-strain fertilization can occur in this species [8].

As far as our knowledge of the genetic control of the life cycle, studies available mostly focus on pennate diatoms. In the pennate benthic diatom *Seminavis robusta*, pheromones mediate the attraction between cells of opposite mating type [9] and determine the transition from mitosis to meiosis, triggering profound changes in gene expression [10]. Chemical cues also play a role in the marine pennate planktonic diatom *Pseudo-nitzschia multistriata*, where sex requires a threshold cell concentration [11] and where transcriptomic experiments provided evidence for a complex cross-talk in the early phases of interaction of the two mating types [12].

The exploration of five diatom genomes and one transcriptome led to the identification of 42 genes potentially involved in meiosis [13]. In this diatom “meiotic toolkit”, five genes are purely meiotic, i.e., they do not have known functions outside meiosis in eukaryotes, and an additional 37 genes are meiosis-related genes, i.e., may have functions in either meiosis or mitosis [13]. Expression data for these meiotic genes in a restricted number of conditions have been collected [13].

Compared to pennate diatoms, relatively little is known on the sexual processes and regulation in centric diatoms. The male gamete of centric diatoms is the only flagellate stage in diatoms life cycles. In the centric diatom *Leptocylindrus danicus*, 22 transcripts related to the synthesis of flagella were shown to be expressed during gametogenesis [14], representing a potential reference dataset to infer sexual events.

With the aim to identify genes linked to the sexual phase of *S. marinoi*, we compared the expression level of transcripts in cells below the SST, which underwent sexual reproduction when exposed to higher salinity [8], with those of large cells above the SST also exposed to higher salinity, which were not competent for sex. We searched for genes related to meiosis [13] and genes related to the formation of flagella [14,15]. We also tested for the presence of genes induced during sexual reproduction in the centric diatom *Thalassiosira weissflogii* [16] and genes putatively linked to the response to sexual cues secreted in the initial phase of sexual reproduction as identified in the pennate diatom *P. multistriata* [12]. We refined the information on the meiotic genes induced during sexual reproduction, and we highlighted unexpected differences in the set of flagellar genes required for the sperm flagella in *S. marinoi* compared to those expressed in the centric diatom *L. danicus* and identified novel sexually induced genes (SIG).

## 2. Materials and Methods

### 2.1. Experimental Setup

The *S. marinoi* strain C1417 was isolated from the germination of resting stages in surface sediments collected in the Gulf of Bothnia (Baltic Sea, 62°07′15″ N, 18°33′14″ E) in April 2011. The strain was deposited at the culture collection of the Finnish Marine Research Infrastructure (FINMARI) of the Finnish Environment Institute and Tvärminne Zoological Station.

For the reference transcriptome (see below), we included reads from *S. marinoi* strain ST54, which originates from a resting cell collected from top layer sediments in Kosterfjord, Sweden (58°51.0 N, 10°45.7 E), in May 2009.

Strain C1417 was grown in 50-mL tissue culture bottles containing 30 mL of f/2 + Si medium [17] prepared with filtered (GF/F) and autoclaved seawater adjusted to a salinity of 6. Standard culture conditions were irradiance of 50 µmol photons m^−2^s^−1^, a photocycle of 14L:10D h, and a temperature of 12 °C.

The sexual phase can be induced in Baltic strains of *S. marinoi* when they are below the SST (i.e., cell diameter ≤6 µm) and upon transfer from the low salinity maintenance condition (salinity 6) to a higher salinity (salinity 16) [8]. A few weeks before the transcriptome experiment, large (diameter >10 µm) and small (diameter ≤5 µm) cells of strain C1417 were isolated to obtain a set of subcultures with large cells (above the SST) and small cells (below the SST), respectively (scanning electron microscopy images in Figure 2A–C). A pilot test was carried out with small cultures of strain C1417 one week before the transcriptome experiment to confirm the timing for the recovery of sexual stages reported in Reference [8]. In this pilot test, observations were carried out after 20, 30, 40, and 62 h after the transfer to media of a higher salinity (Figure 2E). Controls were run with cells above the SST in normal conditions and cells below the SST exposed to higher salinity. The number of sexual stages was not quantified in this experiment, but based on other experiments run under the same conditions, the percentage of auxospores detected at day 3 is estimated to be around 5–9% [8].

The setup for RNA-seq included three experimental conditions: (a) large cells above the SST grown at standard salinity (control condition: no sex, no salinity stress, Figure 2A); (b) large cells above the SST transferred to higher salinity (treatment 1: no sex, salinity stress, Figure 2B); and (c) small cells below the SST transferred to higher salinity (treatment 2: sex, salinity stress, Figure 2C) (Appendix A). Replicate flasks (duplicates for control and treatment 1 and triplicates for treatment 2; see Figure 2A–C) were filled with 300 mL of f/2 medium prepared with oligotrophic seawater adjusted to the required salinity conditions with sterile double distilled water. Cells were dark synchronized for 36 h. The initial cell concentration was at 80,000 cells mL^−1^; bottles were incubated at the standard culture conditions. The sampling points for the RNA-seq experiment were set at 40 h (T1) and 52 h (T2) (Figure 2E).

### 2.2. RNA Extraction and RNA-Seq

Samples were collected onto 1.2-µm pore-size membrane filters (RAWP04700 Millipore), and RNA was extracted with Trizol™ (Invitrogen, Carlsbad, CA, United States) according to the manufacturer’s instructions. The gDNA contamination was removed by DNase I treatment (Qiagen, Hilden, Germany) followed by purification using RNeasy Plant Mini Kit (Qiagen, Hilden, Germany). Quantity of RNA was determined using a Qubit^®^ 2.0 Fluorometer (Life Technologies, Carlsbad, CA, United States) and integrity using a Bioanalyzer (2100 Bioanalyzer Instruments, Agilent Technologies Santa Clara, CA, United States).

Single-end (SE) libraries were prepared using a Beckman Biomek FX and the Illumina^®^ TruSeq^®^ Stranded Total RNA Sample Preparation kit with poly-A selection and starting with 500 ng total RNA. Samples were sequenced on Illumina HiSeq2000 producing SE 50 bp reads. Library preparation and sequencing were done at the Genecore Facility of the European Molecular Biology Laboratory (EMBL), Germany.

In addition to this, in order to facilitate the de novo transcriptome assembly, we also sequenced a strand specific Paired-end (PE) TruSeq^®^ library (2 × 150 bp) on two lanes of an Illumina HiSeq2500 at SciLifeLab Stockholm, Sweden, from a culture of *S. marinoi* ST54.

### 2.3. Transcriptome Assembly and Annotation

Quality control of raw reads was done with the program FastQC (v0.10.1) [18]. Raw SE and PE reads were preprocessed with Cutadapt v1.5 [19] to remove adaptor sequences, as well as any bases with a Phred score below 15. The program fastq_quality_filter v0.0.13 (http://hannonlab.cshl.edu/fastx_toolkit/) was then used to remove sequences with a Phred score below 20 in more than 5% of the bases. A new FastQC run was performed on the filtered data to ensure the validity of preprocessing steps.

The SE reads from the 14 samples, as well as the PE data, were assembled using Trinity v2.1.1 [20] in three different ways: (1) using paired and unpaired reads combined and normalized to a maximum read coverage of 40X, (2) using all available read pairs from the PE dataset, and (3) using only the SE reads normalized to a maximum read coverage of 40X, which resulted in 55,670, 72,727, and 33,004 transcripts, respectively. These 161,401 transcripts where then clustered with a sequence identity threshold of 94% and a word length of 10 bases using cd-hit v4.6 [21], resulting in 48,799 clusters from which one representative sequence was selected. A quality score for each transcript was then calculated using Transrate v1.0.3 [22], which resulted in 26,300 contigs scoring above the optimal assembly score. This analysis was supported by the *Thalassiosira pseudonana* proteome reference v3 ([23]; https://genome.jgi.doe.gov/Thaps3/Thaps3.home.html) and the PE reads used for the assembly. An additional 13,360 contigs below this threshold had a positive contig impact score (reported by Transrate) and were also selected for further analysis. The resulting 39,660 transcripts (Appendix A) were annotated using Annocript v.1.1.3 (https://github.com/frankMusacchia/Annocript) and the UniRef90 database (accessed in May 2019) (Appendix A).

### 2.4. Differential Expression Analysis

Through the Trinity wrappers align_and_estimate_abundance.pl and abundance_estimates_to_matrix.pl [24], SE reads from each biological replicate were individually mapped to the assembled contigs using Bowtie2 v.2.2.7 [25] and the expression of each transcript was quantified using the Expectation–Maximization method (RSEM) [26]. Following the selection of transcripts that had at least one count per million (CPM) in at least 2 samples, the package edgeR 3.12. [27] was used to identify differentially expressed genes (DEGs) for each of the pairwise comparisons at T1 and T2: large cells (>SST) grown in standard condition versus large cells grown at higher salinity (= Salt Stress, SALT) and large cells (>SST) grown at higher salinity versus small cells (<SST) grown at higher salinity (= Salinity shift induced sexual reproduction, SEX) (Figure 2D, Appendix A). Contigs with an FDR < 0.05 and log_2_ fold-change above 0.7 or below −0.7 were considered differentially expressed.

### 2.5. GO Enrichment Analysis

The Gene Ontology (GO) term enrichment analysis was performed on SEX and SALT DEGs (in this case, with −1 > logFC > 1; FDR < 0.05) at T1 and T2. GO term enrichment analysis has been performed using a R plugin of the Annocript software (https://github.com/frankMusacchia/Annocript_utils/blob/master/de_analysis/GO_analysis_4.R), with an adjusted *p*-value cutoff of 0.05 and applying the prop.test function.

### 2.6. Searching for Sex-Related Genes in S. marinoi

To identify putative meiosis, flagella, and sex-induced genes of *S. marinoi*, a multistep procedure of sequence homology analysis was performed. The list of meiotic-related genes was taken from *Arabidopsis thaliana* or *Saccharomyces cerevisiae* [13], the ones for flagella were taken from *Emiliania huxleyi* [15] and *L. danicus* [14], and that for the *T. weissflogii* SIG genes was from Reference [16]. The protein sequences were downloaded from the National Center for Biotechnology Information (NCBI) or retrieved from the Marine Microbial Eukaryote Transcriptome Sequencing Project (MMETSP) datasets (https://www.imicrobe.us/#/search/MMETSP). These proteins were used as query sequences in a BLAST (Basic Local Alignment Search Tool) search, specifically a tBLASTn v2.3.0+ search against the *S. marinoi* transcriptome, where the top five hits with a cutoff E-value of 1E-02 were saved for each gene of interest. The identified transcripts were then compared to the Uniprot–Swissprot database using BLASTx v2.3.0+ [28] to confirm their functional annotation. For the *L. danicus* proteins [14], identity was confirmed with a reciprocal BLAST of the *S. marinoi* top hit against the *L. danicus* transcriptome (MMETSP0321). For the *T. weissflogii* SIG proteins, identity was confirmed with a reciprocal BLAST of the *S. marinoi* hit against the NCBI nonredundant (NR) database.

The 1122 sex-related proteins of *P. multistriata* [12] were used as queries to identify the *S. marinoi* homologs through a tBLASTn search in the whole *S. marinoi* transcriptome (Appendix A). To confirm homology and identity, the best *S. marinoi* hits were used as query in a BLASTx search against the *P. multistriata* gene models and against the NCBI NR database (downloaded in May 2019). For each result, the first top hit with a cutoff E-value of 1E-02 was saved. Finally, common genes between this sex-related homolog dataset and the DEGs dataset were identified (Appendix A).

The *Hap2* and *Gex1* genes, coding for proteins involved in cellular and nuclear fusion, respectively, were suggested as markers for sex in eukaryotes [1]. These genes were searched in the *S. marinoi* reference genome using the Hidden Marcov Model (HMM) approach implemented in HMMER v3.2.1 [29]. Sequences from two Stramenopiles species, *Ectocarpus siliculosus* (CBN75731.1) and Bicosoecid sp., Strain ms1 (CAMPEP_0203808578, CAMPEP_0203813596), were first used as query sequences in a tBLASTn search of the NCBI NR database. The top 15 (HAP2) and 21 (GEX1) BLAST matches were downloaded and aligned together with their respective query sequences using mafft-linsi v7.29b [30]. The alignment result was then examined using the alignment editor SeaView [31], after which sequences deemed to be nonhomologous were manually removed. The final alignments were then used for building an HMM for the respective gene families and subsequently used to search the *S. marinoi* reference genome (https://cemeb.science.gu.se/research/target-species-imago+/skeletonema-marinoi, Töpel et al. in prep.).

### 2.7. Data Deposition

Raw sequence reads were deposited in ArrayExpress (E-MTAB-8091).

## 3. Results

### 3.1. Sexual Reproduction Induced by a Change in Salinity

In cultures of *S. marinoi* above the SST transferred to a higher salinity culture medium (Figure 2C), gametangia, i.e., elongated cells with a bent shape which will produce gametes, were observed at 30, 40, and 62 h and swimming gametes were observed after 62 h (Figure 2E). At the latter observation point, auxospores, i.e., round structures which result from gametes fusion, still attached to the “female” gametangial cells, were also detected. In parallel, we grew large cells above the SST of the same strain. The same salinity treatment applied to these cells (Figure 2B) did not lead to the appearance of any sexual stage since diatom cells below the SST are incapable of performing sexual reproduction. A control culture of large cells not experiencing any change in salinity was also grown (Figure 2A).

### 3.2. De Novo Assembly, Differential Gene Expression Analysis, and GO Enrichment Analysis

To follow gene expression changes linked to the formation of sexual stages, we sampled on day 2 (40 h, T1), when we could clearly detect the presence of gametangia, and on day 3 (52 h, T2). This latter time point was chosen to capture the changes leading to the formation of all the different types of sexual stages (Figure 2E). We expected to observe the induction of genes related to meiosis at both time points, of flagellar genes mostly at T2, and possibly also of genes related to cell–cell signaling linked to the sexual phase that were detected in the pennate diatom *P. multistriata* (likely more pronounced at T1 but maintained at T2 as well) [12].

We sequenced 14 samples, which yielded 169 million SE reads, and an additional 143 million PE reads obtained from an independent single strain were used for the assembly. The de novo transcriptome assembly resulted in 39,660 transcripts that were included in the differential expression analysis. The transcriptome features are reported in Appendix A. Two comparisons were made (Figure 2, Appendix A); the number of DEGs are reported in Figure 3A, and the number of unique and shared DEGs are shown in the Venn diagram in Figure 3B.

Although more genes were globally changing in the comparison between sexualized and non-sexualized cultures (SEX), in the GO term enrichment analysis, few functions were associated to this comparison while the highest number of significantly enriched GO terms was found for the salinity stress alone (SALT) (Appendix A, Appendix A). Specifically, SALT genes displayed more than 4 and 3 times the number of enriched GO terms than SEX genes at T1 and T2, respectively. The enriched functions of the DEGs were higher at T2 in both sets of genes. For the SEX condition, microtubule-related terms were present in all three GO categories (Biological Process, BP; Molecular Function, MF; and Cellular Component, CC) at T1, while terms related to nucleic acid processing and binding (BP and MF) and nucleus (CC) were featured among the enriched GO terms at T2 (Appendix A, Appendix A). For the SALT condition, enrichments in terms linked to metabolic processes (BP and MF) and to organelles such as vacuole and lisosome (CC) were mostly found (Appendix A, Appendix A). A detailed analysis of the genes induced or repressed by the salinity stress alone will be performed in future studies.

### 3.3. Meiosis and Flagellar Genes in the Sexualized Cultures

The putative meiotic proteins listed in Reference [13] were searched in the *S. marinoi* transcriptome, and all the 42 genes of the diatom meiotic toolkit were detected. We could also detect the presence of *Brca1* (breast cancer type 1 susceptibility protein), a gene required for DNA repair in mitosis and meiosis, not reported previously [13]. A total of 13 and 30 meiotic-related genes were found upregulated in sexualized cultures at T1 and T2, respectively, with only two genes, *Smc2* and *Smc4* (structural maintenance of chromosomes protein 2 and 4, two proteins forming a heterodimer required for chromosome condensation) being upregulated at T1 but not at T2 (Table 1).

Among the DEGs at both time points were *Spo11-2*, *Msh4*, and *Msh5*, known to be expressed exclusively during meiosis (Table 1). Interestingly, the meiosis-related genes *Mnd1*, *Mer3*, *Mcm3*, *Mcm7*, *Mcm9*, *Smc6*, *Rad21*, *Rad52*, *Rad1*, *Msh2*, *Fancm*, and *Fen1*, which are part of the group of genes upregulated during sexual reproduction in the pennate diatom *S. robusta* [13], were not differentially expressed in our transcriptomic dataset.

A search was made also for HAP2 (GCS1) and GEX1 (KAR5), two proteins known to be almost ubiquitously required in gamete fusion and karyogamy, respectively [1]. No homology was found for HAP2 in the *S. marinoi* genome while for GEX1 the search retrieved one region, corresponding to the gene Sm_00029435-RA, with a low similarity to the query sequence. However, a reciprocal BLAST of the *S. marinoi* sequence in the NCBI NR database retrieved a different best hit, not confirming Sm_00029435-RA as a GEX1 homolog.

Flagellate gametes had been seen in the sexually reproducing culture, and we thus searched for genes known to be required for flagellar assembly and intraflagellar transport (IFT). A set of genes expressed in the flagellate haploid stage of the prymnesiophycean *E. huxleyi* is available [15,32], and IFT genes have been recently reported also for the centric diatom *L. danicus* [14]. A total of 19 flagellar genes expressed in *E. huxleyi* appeared to be differentially regulated in *S. marinoi*, with the majority of changes recorded at T2 (18 DE genes versus 6 DE genes in T1), consistent with the appearance of flagellate gametes after 40 h (Figure 1, Table 2).

Of the differentially expressed flagellar genes, many were related to dyneins, specific to flagella, while others are known to have other functions in the cytoplasm. Surprisingly, none of the 22 *L. danicus* flagellar genes was found to be differentially expressed in *S. marinoi*. In fact, only eight out of the 22 *L. danicus* flagellar genes found homologues in the transcriptome; the remaining 14 genes did not find any homology (Appendix A). To verify if the lack of flagellar gene expression could be due to a weak signal, i.e., the limited number of male gametes produced, we tested how many of the missing *L. danicus* flagellar genes were present in the *S. marinoi* draft genome sequence. We could confirm the presence of the eight genes found in the transcriptome and could retrieve two more genes of the 14 missing (Appendix A), bringing the total number of homologues present in *S. marinoi* to 10.

### 3.4. Homologies with Sex-Related Genes of the Pennate Diatom Pseudo-nitzschia multistriata and of the Centric Diatom Thalassiosira weissflogii

We tested the presence in the *S. marinoi* transcriptome of homologues of the 1122 genes that have been found to be regulated during the early stages phases of sexual reproduction in the pennate diatom *P. multistriata* [12]. A total of 592 homologous transcripts were identified; 179 of them were DEGs in *S. marinoi*, of which 24 were also responding to the salinity shift (Appendix A). Of the 155 regulated only in the SEX condition, 17 transcripts were upregulated and 9 were downregulated at both time points in *S. marinoi* and, with the exception of three transcripts (*P. multistriata* logFC marked with “a” in Table 3), they all changed in the same direction as the *P. multistriata* regulated genes (Table 3).

Three sex induced genes (*Sig1*, *Sig2*, and *Sig3*) reported for the centric diatom *T. weissflogii* [16] were searched in the transcriptome. One hit was found for *Sig1*, and two hits were found for *Sig2* and *Sig3* (Appendix A). All transcripts were upregulated at T2; *Sig1*, one of the *Sig2* homologous transcripts and one of the *Sig3* homologous transcripts were also upregulated at T1.

### 3.5. Other Major Transcriptomic Changes

Most of the top upregulated transcripts were represented by proteins that had not yet received any annotation. Among the few that could be annotated and were present at both time points in the SEX but not in the SALT condition, we found a putative phosphatidylinositol phosphate kinase (STRINITY_DN15608_c0_g1_i1) with a LogFC of ~11 at both time points and a MAP kinase (Pc16572_g1_i4) with 6.7 and 9.8 logFC at T1 and T2 respectively. A Leucine-rich repeats (LRR) domain containing protein (STRINITY_DN12542_c0_g2_i2) with a logFC of 10.5 at T1 and 7.5 at T2 is interestingly homologous to numerous proteins in the centric diatom *Thalassiosira oceanica* but does not find similarities in any other diatom. Other strongly upregulated genes were an HMG (high mobility group) box domain containing protein (Pc29422_g1_i2), known to be important for chromatin structure and dynamics, with 9.7 and 6.2 logFC at T1 and T2, respectively, and a Predicted ATPase, AAA+ superfamily domain containing protein (STRINITY_DN21065_c0_g1_i1) with 7.8 and 10 logFC at T1 and T2, respectively. More transcripts with functions related to signal transduction and DNA dynamics could be found strongly upregulated (Appendix A). As far as functions that appeared to be downregulated, we noted that the majority of the transcripts that had logFC < −3 at T2 (151 genes) were putatively localized in the chloroplast (Appendix A), while this was not observed at T1, where only 12 genes had logFC < −3 and none was chloroplastic.

## 4. Discussion

Centric diatoms dominate planktonic communities in many regions of the world oceans [33]. The importance of their life cycles, which impact population dynamics, motivates the need to gain a better understanding of the triggers of life cycle transitions and of the genetic programmes involved in their control. For *S. marinoi*, a change in salinity is a reliable cue triggering sexual reproduction [8]. A transcriptomic approach was used to identify the major pathways associated with the switch from the vegetative lifestyle to sexual reproduction, with the aim to highlight similarities and differences with other known systems.

As expected, one of the strongest signals detected in the transcriptome of the sexualized diatom *S. marinoi* was the upregulation of genes involved in meiosis. Surprisingly, despite the fact that flagellate gametes were observed in the sample, we did not observe changes in any of the genes involved in the synthesis of the flagella that were upregulated during sexual reproduction in another centric diatom, *L. danicus* [14]. Actually, many of these genes were not found in the *S. marinoi* genome at all. Moreover, a comparison with the set of genes changing during the first phases of sexual reproduction of the pennate diatom *P. multistriata* revealed the existence of commonalities, including the strong upregulation of genes with an unknown function that we suggest to name Sex Induced Genes (SIG), in analogy to what has been proposed for the centric diatom *T. weissflogii* [16].

### 4.1. Meiosis-Related Genes

We selected two different time points during the process of sexual reproduction. More meiotic genes were differentially expressed at the second time point as compared to the first one. This can be explained by the fact that T2 was a more advanced phase of sexualization of *S. marinoi* where more types of sexual stages were present as compared to T1 where only female gametangia were clearly visible. Among the genes that are exclusively required for meiosis, *Spo11-2* was recorded at both T1 and T2 with extremely high fold changes (11.9 and 8.4 logFC, respectively). This protein catalyzes meiotic DNA double strand breaks (DSB) in homologous chromosomes during meiotic recombination. In the diatom genomes examined by Reference [13], four additional meiotic-specific genes were reported, *Msh4*, *Msh5*, *Mer3*, and *Mnd1*, which were present also in the genome of *S. marinoi*, although only the first two genes were differentially expressed in the sexualized samples. All five meiosis-specific genes were upregulated in the transcriptome from cocultures of the pennate diatom *S. robusta* undergoing meiosis [13]. *Msh4* and *Msh5* are involved in the formation of chiasma [34], and *Mer3* is reported to promote DNA recombination together with *Msh4* and *Msh5* [35]. *Mnd1* mediates homologous DNA pairing during meiosis together with *Hop2* [36], a gene which is absent in diatoms ([13] and this study). *Hop2* and *Mnd1* have been identified in yeast as factors supporting homologous chromosome pairing and meiotic DSB repair [37]. However, in some model organisms such as the nematode *Caenorhabditis elegans* and the fruit fly *Drosophila melanogaster*, meiosis can proceed via a recombination-independent matching and locking of homologous chromosomes in the absence of *Hop2*, *Mnd1*, and *Dmc1* [38]. The same mechanism might apply to *S. marinoi*, in which *Hop2* and *Dmc1* are missing, while *Mnd1*, although present, is not induced during sexual reproduction.

Besides the genes exclusively linked to meiosis, an increased fold change was detected in sexualized *S. marinoi* also for 25 genes that, in diatoms, are linked to meiosis but have other roles also in mitosis. These genes have been shown to be all differentially expressed during the sexual phase of *S. robusta* [13], thus supporting the hypothesis that they are involved in meiotic events in both centric and pennate diatoms.

Together with *Spo11-2*, *Rad51-A* represents one of the earliest and strongest signals detected, confirming data in *P. multistriata* and *S. robusta* [13]. Both genes, therefore, represent good markers for meiosis detection, although their being part of multigene families [13] requires gene phylogeny analyses to support proper identification.

As far as the other genes expected to be induced, based on the *S. robusta* data [10,13] but not regulated in the dataset of *S. marinoi*, one explanation could be that their expression levels were already sufficient to allow for the meiotic machinery to function during the progression of sexual reproduction.

### 4.2. Flagellar Genes

The eukaryotic flagellum is an ancient structure, common in protists and in many multicellular organisms, and it is believed that its absence in some groups is due to a secondary loss [39]. While pennate diatoms have completely lost flagella, centric diatoms retain it as a transient structure in their male sperms [6].

We do not have a complete knowledge of the ultrastructure and composition of diatom flagella. The typical 9 + 2 flagellar axoneme is present in *Bolidomonas*, an ancestor of the diatom lineage [40]. No information on the ultrastructure of the male gamete flagella of *S. marinoi* or *L. danicus* is available, while there are four studies reporting data for *Lithodesmium undulatum* [41], *Pleurosira laevis* (as “*Biddulphia levis*” in Reference [42]), *Coscinodiscus wailesii* and *Chaetoceros laciniosus* [43], and *Thalassiosira lacustris* and *Melosira moniliformis* var. *octogona* [44]. In the studied species, the main differences from the typical flagellum in heterokonts are the presence of axonemes with a 9 + 0 configuration, lacking the central pair of microtubules and the radial spoke; the lack of microtubular roots and rhizoplast; and the presence of basal bodies constituted by nine microtubule doublets rather than nine triplets in species from the genera *Melosira*, *Thalassiosira*, *Pleurosira*, and *Coscinodiscus* [44].

As far as genes required for flagella assembly, in the genome of the centric diatom *T. pseudonana*, genes encoding IFT proteins (used for translocation of large protein complexes along microtubules [45]) were found for the complex B subunits, along with genes for the kinesin II motor that transports proteins to the ciliary tip [46]. However, no components of heavy and light-intermediate chains of cytoplasmic dynein 2⁄1b or IFT complex A subunits were recorded. The IFTA gene *Ift140* was, however, found in the transcriptomes of five centric diatoms, *T. weissflogii*, *L. danicus*, *Leptocylindrus hargravesii*, *Rhizosolenia setigera*, and *Proboscia alata* [14], while another IFTA gene, *Ift122*, was found in the same species as well as in the transcriptomes of *Stephanopyxis turris*, *Aulacoseira subarctica*, *Ditylum brightwellii*, and *Odontella aurita*. Various IFTB genes were found in many radial and multipolar centric diatoms [14].

When looking for genes related to the formation and maintenance of flagella in *S. marinoi*, we found genes encoding dyneins and other cytoskeleton proteins that are part of flagella, which are expressed in the flagellate haploid life cycle stage of *E. huxleyi* [15]. We did not, however, find any overlap with the set of flagellar genes expressed in *L. danicus* sexually reproducing cultures [14]. Missing genes include key flagellar genes such as *Ift122* and *Ift140* (IFTA) or *Ift172* and *Ift80* (IFTB).

One hypothesis is that a broader set of flagellar genes is present in *L. danicus* and *L. hargravesii,* which belong to an evolutionary ancient genus at the base of diatom phylogeny, and that the number of flagellar genes gradually reduces in more recent centric diatoms such as *S. marinoi*. It has to be considered that flagella are present only in a very limited time interval of the life cycle of centric diatoms and that it is, thus, possible that the number of genes has been reduced over the evolutionary history, along with the presence of a simplified ultrastructure of the flagellum (see above). Functional and structural modularity within the IFT complex has been reported, and the gain and loss of IFT components in different groups have been proposed to occur in distinct modular steps [45], in line with our observations in diatoms.

It is also interesting to note that the transcriptome data of *L. danicus* were obtained from a culture in exponential growth phase, grown in a full-strength culture medium and not in a nitrogen depleted medium known to elicit the sexual phase [14,47]. The complete set of flagellar genes reported as upregulated in the transcriptome of *L. danicus* can be taken as proof that sexual reproduction must have occurred when the culture was grown in standard conditions. Similarly, the finding of flagellar genes in the MMETSP transcriptomes of several centric diatoms [14] suggests that sex may occur more frequently than expected and that these species do not require specific environmental cues to undergo the sexual phase.

### 4.3. Other Sex Related Genes

Some of the most induced genes were not annotated, and we propose to name those that have a logFC ≥ 3 in one of the two time points and are induced in *P. multistriata* as well as SIG (Sex Induced Gene) (Table 3) in analogy to what has been done for three genes found upregulated in the diatom *T. weissflogii* during sexual reproduction [16]. The three SIG proteins in *T. weissflogii* share conserved domains, which are probably involved in cell–cell interactions, and were hypothesized to be required for egg–sperm interactions [16]. Moreover, homologues of the *Sig1* gene identified by Reference [16] were shown to code for proteins of the mastigonemes, the tripartite tubular hairs found on the stramenopile flagellum [48]. It has to be noted that the designation we propose for *S. marinoi* genes is not related to a structural homology between the genes but is based exclusively on their expression profile. The addition of novel datasets from other diatom species undergoing sexual reproduction will reveal the extent of conservation of these genes and the specificity for the process under investigation. It is possible that some of these genes in *S. marinoi* are required for pheromone production or sensing. Pheromones must be species-specific to avoid breeding between cells of different species; however, processes downstream of the response to the very first signals might be conserved across species.

Other genes worthy of investigation are the ones highly induced and annotated as signal transduction proteins. One of the responses to sexual cues reported for pennate diatoms is a general arrest of cell growth [10,11]. *S. marinoi* cells, however, do not seem to experience a marked growth arrest, and coherently, we did not find major signals related to the cell cycle control. It is interesting to note a general downregulation of genes related to the chloroplast function. A similar observation (downregulation of genes linked to photosynthesis) was made for cells collected during the late phase of sexual reproduction in *P. multistriata* (MI Ferrante, Personal communication). The significance of these observations remains to be explored with dedicated experiments.

We believe that a great part of the changes in our SEX comparison is due to the major change occurring in the cell population (the switch from mitosis to meiosis), as confirmed by the fact that the number of DEGs increases at the second time point when more sexual stages have been produced with respect to the first time point. It is important to mention, however, that with our experimental setup, where we compared large cells with the salinity shock with small cells with the salinity shock, we cannot exclude that some of the general changes observed (such as those described in Section 3.5) could be due to the different gene expression profiles of large versus small cells, independently of sexual reproduction. A dedicated experiment to compare the gene expression changes linked to variation in the cell size (cells above the SST versus cells below the SST) in standard growing conditions will be needed to refine our knowledge of the process.

The availability of genomic and transcriptomic resources has allowed comparative studies amongst different species with indications of a set of genes related to key phases in the life cycle of unicellular organisms. This is the case of the “meiosis” toolkit [13,49] discussed above and other potential markers of sexual reproduction in eukaryotes, such as the HAP2 and GEX1 proteins involved in cell and nuclear fusion [1]. Homologues of these proteins were not found in the transcriptome and the genome of *S. marinoi* and in the genome of the pennate diatom *P. multistriata* (MI Ferrante personal communication). Among Stramenopiles, both transcripts were reported only for an unidentified Bicosoecid (Bicosoecid sp. Strain ms1), while only *Gex1* was reported in the brown macroalga *E. siliculosus* and in other not specified taxa [1]. A broader sampling of the stramenopile lineage is thus required to depict the evolutionary history of HAP2 and GEX1 proteins and, more in general, the evolution of sexual recognition systems that allow the haploid gamete to recognize the complementary haploid cell within the same species and then to fuse with it [2].

## 5. Conclusions

The growing interest in the functioning of ocean ecosystems, the awareness of the role of unicellular organisms at the base of marine food webs, and the increasing availability of genomic and transcriptomic tools is opening a new era in our understanding of the biology and ecology of these organisms. Our results contribute to the identification of a “sex toolkit”, including *Spo11-2*, *Rad51-A*, and possibly SIG genes, that will be useful to spot sexual reproduction events of diatoms in the natural environment through the interpretation of metagenomics and metatranscriptomics data. Many of the genes that turned out to be upregulated during sexual reproduction are not annotated, but it is through the comparison of datasets obtained for different species that we will be able to identify the genes involved in distinct phases of the diatom life cycle, the genomic networks in which they are involved, and their modifications during the diatom evolutionary history.

## Figures and Tables

**Figure 1 genes-10-00494-f001:**
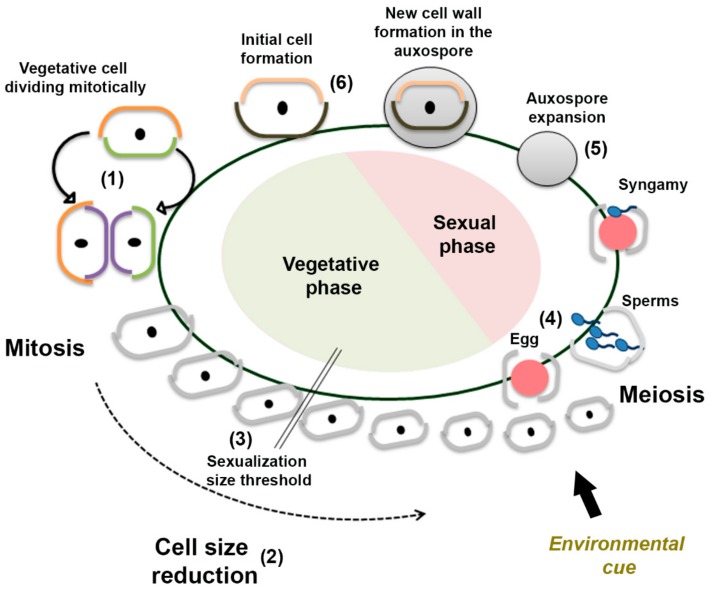
Schematic drawing of the life cycle of a centric diatom: When a vegetative cell divides mitotically (**1**), each daughter cell inherits one of the two halves (thecae) of the rigid silica cell wall (orange and green lines) and builds a new smaller theca (purple lines). This leads to a progressive reduction of the cell size of the cell population (**2**). Above a species-specific sexualization size threshold (SST), (**3**) diatom cells are incapable of performing sexual reproduction. Below the SST, if a proper trigger is present (such as a salinity shock for *Skeletonema marinoi*), meiosis is induced and cells can either produce one egg or four sperms (**4**). Gametes conjugation (syngamy) leads to the formation of a zygote that becomes an auxospore, a soft stage which can expand (**5**). A new cell wall is built inside the auxospore, which eventually becomes an initial cell of the maximum cell size (**6**).

**Figure 2 genes-10-00494-f002:**
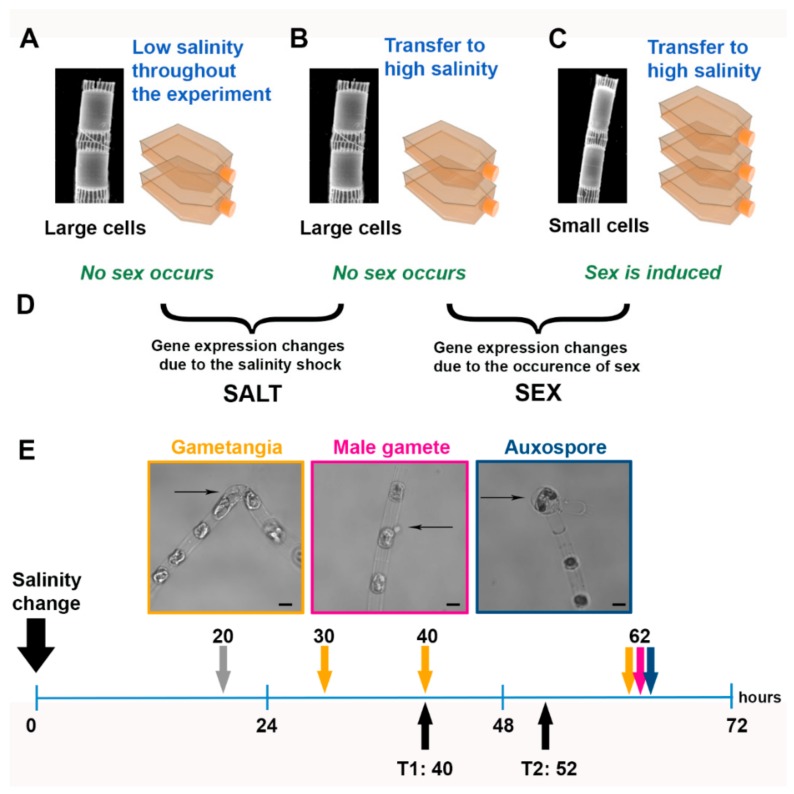
Experimental setup: (**A**–**C**), conditions tested in the experiment. The scanning electron micrographs show two *S. marinoi* cells in a chain. (**D**) Comparisons considered for the gene expression analysis. (**E**) Scheme of the sexual stages of *S. marinoi* detected at different time points after application of the salinity cue (0 h) (see Reference [8]) and time points chosen for the transcriptomic analysis (black arrows). At 20 h (gray arrow), only vegetative cells were observed. The yellow arrows, the pink arrow, and the blue arrow indicate the presence of gametangia, gametes, and auxospores, respectively. The sexual stages, indicated by thin black arrows, are shown in the images above with a color-matching frame. Scale bar, 5 µm.

**Figure 3 genes-10-00494-f003:**
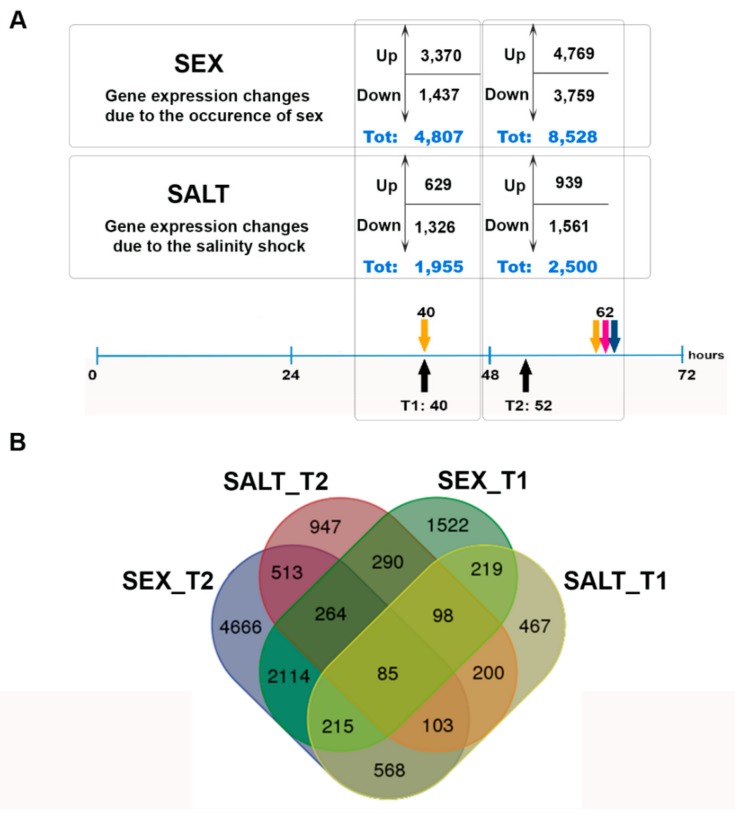
Number of genes differentially expressed in the SEX and SALT comparisons: (**A**) Number of genes up- and downregulated at the two time points. (**B**) Venn diagram showing the overlap between different datasets of differentially expressed genes in the SEX and SALT conditions at the two time points.

**Table 1 genes-10-00494-t001:** Differentially expressed contigs of *S. marinoi* (−1 > logFC > 1) with significant BLAST hits to meiosis genes.

Function	Protein Name	Query ID (a)	*S. marinoi* Hit ID	Hit ID Annotation	SEX_T1 logFC	SEX_T2 logFC	SALT_T1 logFC	SALT_T2 logFC
Accessory proteins required during meiosis	BRCA1	AAO39850.1	MTRINITY_DN15891_c0_g1_i1	Protein breast cancer susceptibility 1 homolog	-	1.54	-	-
	BRCA2 (b)	AEE81814.1	MTRINITY_DN15273_c1_g3_i3	Breast cancer type 2 susceptibility protein homolog	-	1.65	-	-
	DNA2 (b)	NP_001184943.1	MTRINITY_DN16883_c0_g1_i2	DNA replication ATP-dependent helicase/nuclease JHS1	-	1.51	-	-
	EXO1 (b)	Q8L6Z7.2	Pc10978_g1_i1	Exonuclease 1	-	7.18	-	-
Crossover regulation	MSH4 (b)	AAT70180.1	Pc32435_g1_i6	DNA mismatch repair protein MSH4	4.44	6.64	-	-
	MSH5 (b)	NP_188683.3	Pc31899_g1_i1	MutS protein homolog 5	2.76	1.95	-	-
DNA damage sensing and response	RAD50 (b)	AEC08614.1	MTRINITY_DN15782_c0_g2_i2	DNA repair protein RAD50	-	2.89	-	-
	MRE11 (b)	NP_200237.1	Pc25688_g1_i2	Double-strand break repair protein MRE11	-	1.74	-	-
DNA replication and chromosome maintenance	SCC3 (b)	AEC10920.1	MTRINITY_DN16741_c0_g1_i1	Cohesin subunit SA-3	-	3.45	-	-
	SMC5 (b)	AED92224.1	Pc30152_g1_i1	Structural maintenance of chromosomes protein 5	-	2.24	-	-
	MCM6	AED95141.1	MTRINITY_DN14495_c0_g1_i2	DNA replication licensing factor MCM6	-	1.84	-	-
	SMC4 (b)	AED95695.1	MTRINITY_DN13431_c0_g1_i1	Structural maintenance of chromosomes protein 4	3.24	-	−1.59	−1.27
	SMC1 (b)	AEE79265.1	Pc27448_g3_i1	Structural maintenance of chromosomes protein 1	-	2.71	-	-
	MCM2 (b)	NP_001185154.1	Pc31350_g2_i1	DNA replication licensing factor MCM2	-	1.66	1.17	-
	MCM5 (b)	NP_001189521.1	MTRINITY_DN18289_c0_g1_i1	DNA replication licensing factor MCM5	-	2.16	-	-
	PDS5 (b)	NP_177883.5	MTRINITY_DN17063_c0_g1_i3	Sister chromatid cohesion protein PDS5 homolog A	1.16	2.51	-	-
	MCM4	NP_179236.3	MTRINITY_DN16417_c4_g1_i2	DNA replication licensing factor MCM4	-	2.43	-	-
	SMC3 (b)	NP_180285.4	MTRINITY_DN4029_c0_g2_i1	Structural maintenance of chromosomes protein 3	1.06	3.00	-	-
	MCM8 (b)	NP_187577.1	MTRINITY_DN16028_c0_g1_i1	Probable DNA helicase MCM8	-	1.13	-	-
	SMC2 (b)	NP_201047.1	Pc22708_g1_i1	Structural maintenance of chromosomes protein 2-1	3.33	-	−1.41	−1.31
Double-strand break formation	SPO11-2 (b)	AEE34178.1	MTRINITY_DN15680_c0_g1_i1	Meiotic recombination protein SPO11-2	11.92	8.42	-	-
	SPO11-3/TOPVI	NP_195902.1	Pc22702_g1_i1	Topoisomerase 6 subunit A3	-	1.58	-	-
Double-strand break repair (recombinational repair)	PMS1 (b)	AAM00563.1	Pc31460_g3_i1	DNA mismatch repair protein PMS1		1.38	-	-
	RAD51-A (b)	BAE99388.1	STRINITY_DN8677_c0_g1_i2	DNA repair protein RAD51 homolog 1	4.10	2.28	-	−1.52
	RAD51-C (b)	CAC14091.1	Pc29686_g1_i1	DNA repair protein RAD51 homolog 3	-	1.43	-	-
	MSH6 (b)	NP_001190656.1	MTRINITY_DN17310_c0_g1_i2	DNA mismatch repair protein Msh6	1.36	2.10	-	-
	RAD51-B (b)	NP_180423.3	MTRINITY_DN12845_c1_g1_i3	DNA repair and recombination protein RadA	-	1.29	-	-
	XRCC3	NP_200554.1	Pc31108_g1_i2	DNA repair protein XRCC3 homolog	1.54	1.23	-	-
	MLH1 (b)	NP_567345.2	MTRINITY_DN12931_c0_g1_i1	DNA mismatch repair protein MLH1	-	1.56	-	-

(a) All queries are *Arabidopsis thaliana* proteins except PMS1 which is from *Saccharomyces cerevisiae*; (b) genes that were upregulated during meiosis in the diatom *Seminavis robusta* [13].

**Table 2 genes-10-00494-t002:** Differentially expressed contigs of *S. marinoi* (−1 > logFC > 1) with significant BLAST hits to flagellar genes.

Uniprot Description	Uniprot ID	*S. marinoi* Hit ID	Hit ID Annotation	SEX_T1 logFC	SEX_T2 logFC	SALT_T1 logFC	SALT_T2 logFC
Dynein β chain, flagellar outer arm	DYHB_CHLRE	MTRINITY_DN17388_c0_g1_i3	Dynein β chain flagellar outer arm	-	3.58	-	-
Dynein, 70 kDa intermediate chain, flagellar outer arm	DYI3_CHLRE	STRINITY_DN18690_c0_g1_i1	Dynein 70 kDa intermediate chain flagellar outer arm	-	8.40	-	-
Dynein heavy chain	Q5H9M7_BOVIN	Pc32863_g3_i1	Dynein heavy chain 8 axonemal	4.39	4.61	-	-
Dynein light chain roadblock	Q7ZUX8_DANRE	STRINITY_DN2071_c0_g1_i1	Dynein light chain roadblock-type 1	-	8.55	-	-
Tctex1 domain-containing protein 1-B	TC1DB_XENLA	MTRINITY_DN20279_c0_g1_i1	Tctex1 domain-containing protein 1	-	7.64	-	-
Nucleoside diphosphate kinase 7	NDK7_RAT	STRINITY_DN12961_c0_g1_i1	Nucleoside diphosphate kinase 7	-	2.38	-	-
Dynein light chain Tctex-type 1	DYLT1_MOUSE	STRINITY_DN18525_c0_g1_i1	Dynein light chain Tctex-type 1	-	7.05	-	-
Uracil phosphoribosyltransferase	UPP_ROSS1	MTRINITY_DN17186_c0_g1_i2	ATP-dependent D- helicase Q-like 5	-	1.95	-	-
EF-hand domain-containing protein 1	EFHC1_MOUSE	STRINITY_DN11650_c0_g1_i2	EF-hand domain-containing family member C2	2.49	4.56	-	-
Dynein regulatory complex subunit 4	GAS8_CHLRE	STRINITY_DN15037_c0_g1_i1	Dynein regulatory complex subunit 4	5.38	7.09	-	-
Dynein regulatory complex subunit 2	CCD65_BOVIN	STRINITY_DN9819_c0_g1_i1	Coiled-coil-domain-containing-protein-65	9.26	8.97	-	-
Tubulin α-2 chain	TBA2_PELFA	STRINITY_DN11650_c0_g2_i3	Tubulin α-2 chain	-	2.16	-	−1.56
Tubulin β chain	TBB_TETTH	STRINITY_DN3540_c0_g1_i1	Tubulin β chain	6.63	8.51	-	-
Actin. non-muscle 6.2	ACT_HYDVU	STRINITY_DN3928_c0_g1_i1	Actin	-	0.98	-	−1.17
Caltractin	CATR_SCHDU	MTRINITY_DN2103_c0_g1_i1	Caltractin	-	1.89	-	-
Glycogen synthase kinase-3 α	GSK3A_RAT	MTRINITY_DN14448_c0_g1_i1	Glycogen synthase kinase-3 β	-	1.18	-	-
Microtubule-associated protein RP/EB family member 1	MARE1_XENTR	Pc31703_g4_i1	Microtubule-associated protein RP/EB family member 1B	1.40	-	-	−1.47
Serine/threonine-protein phosphatase 4 regulatory subunit 1	PP4R1_RAT	MTRINITY_DN15306_c0_g1_i1	Serine/threonine-protein-phosphatase-4-regulatory-subunit-1	-	-1.56	-	-

**Table 3 genes-10-00494-t003:** Differentially expressed contigs of *S. marinoi* with significant BLAST hits to sex related genes of *P. multistriata* [12] common at the two time points.

*S. marinoi* Contig Id	Contig Annotation	SEX_T1 logFC	SEX_T2 logFC	*P. multistriata* Homolog	*P. multistriata* Annotation	*P. multistriata logFC*	Proposed Gene Name
MTRINITY_DN14053_c0_g1_i4	-	8.46	5.68	0002850.1	-	4.03	*SIG4*
MTRINITY_DN1615_c0_g1_i1	-	6.96	6.35	0059370.1	-	2.92	*SIG5*
MTRINITY_DN10312_c0_g1_i1	Multidrug resistance protein 1	5.85	2.67	0056660.1	ABC transporter B family member 3	2.14	
Pc26735_g2_i3	-	5.70	3.53	0079640.1	-	5.85	*SIG6*
STRINITY_DN12692_c0_g1_i1	-	5.68	5.70	0010180.1	-	1.85	*SIG7*
STRINITY_DN10082_c0_g1_i1	-	4.73	6.47	0055440.1	-	1.74	*SIG8*
Pc19236_g1_i3	Anaphase-promoting complex subunit cdc20	4.49	2.32	0095480.1	Anaphase-promoting complex subunit cdc20	−1.44 (a)	
Pc15065_g1_i1	-	4.10	2.19	0009930.1	-	4.89	*SIG9*
MTRINITY_DN9343_c0_g1_i2	-	3.42	2.68	0061540.1	-	2.47	*SIG10*
Pc26832_g1_i1	-	3.23	3.50	0093610.1	-	4.12	*SIG11*
MTRINITY_DN8414_c0_g2_i1	-	2.09	1.69	0076960.1	Dynamin-like protein A	−1.70 (a)	
MTRINITY_DN15107_c0_g1_i1	-	1.67	3.14	0000630.1	-	−1.91 (a)	
MTRINITY_DN17063_c0_g1_i3	Sister chromatid cohesion protein PDS5 homolog A	1.16	2.51	0089060.1	-	1.95	
STRINITY_DN5368_c0_g1_i1	-	1.12	2.95	0100360.1	-	1.52	
MTRINITY_DN4029_c0_g2_i1	Structural maintenance of chromosomes protein 3	1.06	3.00	0079810.1	Structural maintenance of chromosomes protein 3	1.67	
Pc28241_g1_i1	mRNA decay activator protein ZFP36L1	0.98	1.39	0076260.1	-	3.70	
Pc29361_g2_i1	-	1.46	2.68	0074970.1	-	1.47	
MTRINITY_DN12258_c1_g1_i2	Translation initiation factor IF-2	−2.05	−2.10	0047520.1	Translation initiation factor IF-2	−2.14	
MTRINITY_DN7716_c0_g1_i1	Putative elongation factor TypA-like SVR3	−1.46	−0.87	0044870.1	GTP-binding protein TypA/BipA homolog	−1.37	
MTRINITY_DN17106_c0_g1_i1	Acetyl-CoA carboxylase 1	−1.21	−0.77	0084140.1	Acetyl-CoA carboxylase 1	−1.86	
MTRINITY_DN12117_c0_g1_i1	Leucine-tRNA ligase	−1.06	−1.05	0031170.1	Leucine--tRNA ligase	−1.60	
STRINITY_DN8561_c3_g1_i1	-	−1.00	−1.39	0040850.1	Glucose-repressible alcohol dehydrogese transcriptional effector	−2.13	
STRINITY_DN13193_c0_g1_i1	Pleckstrin homology domain-containing family A member 8	−0.94	−1.11	0005620.1	-	−1.81	
MTRINITY_DN16739_c0_g1_i5	Bifunctional purine biosynthetic protein ADE1	−0.80	−1.11	0025170.1	Phosphoribosylformylglycimidine cyclonaligase	−1.28	
MTRINITY_DN9740_c0_g1_i1	Arogenate dehydrogenase 2 chloroplastic	−0.78	−1.03	0020750.1	Arogenate dehydrogenase 2 chloroplastic	−2.34	
MTRINITY_DN17191_c0_g2_i1	Ribosomal RNA processing protein 1 homolog B	−0.78	−1.61	0119100.1	-	−1.99	

(a) LogFC values not in accordance with the direction of change in *S. marinoi*.

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
