# Peer review of "Exploring Molecular Signs of Sex in the Marine Diatom Skeletonema marinoi"

_genes, 2019, doi:10.3390/genes10070494_

Round 1

Reviewer 1 Report

Ferrante et al investigate the modification of the expression profile of the diatom Skeletonema marinoi occurring during sexual reproduction in order to identify genetic markers of sexual reproduction. To identify these markers, the authors induced sexual reproduction by applying a salt stress to big cells that are capable of sexual reproduction. As controls, they also investigated the expression profiles of small cells incapable of sexual reproduction when a salt stress is applied, and of big cells without the salt stress. These expression profiles were obtained at two time points. The discussion of the results is extremely well documented.

My main comments relate to the comparison of the different treatments. First, I had to spend quite some time and to keep moving from the Materials and Methods to the Results section in order to understand which comparisons you were writing about. Maybe you could use a graph to illustrate your experimental design. Second the SS and Sex denomination is quite confusing. Indeed the SS comparison compares sex vs no sex in big cells but with salt stress as a confounding factor, while the Sex comparison compares sex vs no sex without salt stress but with cell size as a confounding factor. So why do you think the Sex comparison is more informative than the SS comparison (cf L147-148, 167-168, Result section)? You should rather deal with the confounding factors by either: 1. Trying to Introduce the full experimental design in the statistical model and performing a single DE analysis at each time point, or 2. Performing several statistical analyses (as you did) but considering as good candidates the transcripts DE at the intersection between the SS and Sex comparisons. Third, why do you put emphasis on the transcripts DE in T1 and T2 (intersection)? The cells are not supposed to be in different cell stages at these two time points (see comment below)?  

 Are your cultures synchronous regarding sexual reproduction? Is there a way to know what fraction of the population is actually involved in sexual reproduction at your sampling time points? How do you think this might influence your results?

L74: I would not say that L. danicus is ancestral but rather that it diverged ancestrally

L120: please explain how you end up with 14 samples. To me you have 3 biological conditions (small with salt stress, big with salt stress and big without salt stress)*2 time points (40 and 52 h) = 6 different conditions. If you have two replicates (which is a very low level of replication) you end up with 12 samples.

L126: How did you select your representative sequences? Moreover, I am not sure the 94% identity threshold is a good way to deal with isoforms. How do you deal with isoforms ?

L191: change > to <

L177: why 15 and 21 ?

L203-204: Please explain why you chose these two time points

L111-112 and 210: please explain why you used this different strain?

 Tables 2-4:  You select your DE transcripts using a -1<logFC<1 and FDR<0.05 threshold (mat and meth), so I would either remove the “-1<logFC<1” or add “FDR<0.05” in table 2,3 and 4.     

L274-287: I would add a graph (venn or anything else) illustrating these results.  

Author Response

REFEREE #4

Comments and Suggestions for Authors

This study explores genes that are expressed in a diatom during growth conditions that support formation of gametes and sexual reproduction.   Genes associated with the meiotic toolkit were found to be expressed, whereas flagellar genes were not found to be expressed.  The first result is consistent with sexual reproduction, whereas the later result is somewhat unexpected given results from previous studies of other species.  This study is fine as far as it goes and may be what is possible in this system/organism.  What would make this of greater impact is if the introduction and discussion covered advances in what is known about other stremenopiles such as oomycetes, or other organisms such as fungi and algae.  The authors could consider for example citing a recent chapter by Goodenough and Heitman that covers an evolutionary perspective on sexual reproduction:

Origins of eukaryotic sexual reproduction.

Goodenough U, Heitman J.

Cold Spring Harb Perspect Biol. 2014 Mar 1;6(3). pii: a016154. doi: 10.1101/cshperspect.a016154. Review.

Thank you for the positive comments.

We have added a new paragraph at the beginning of the introduction and a sentence in the discussion with reference to the suggested review.  

Specific questions.

Has a mating type locus or sex chromosome been identified in this or any other diatom?

So far, there are two reports only that address this issue, both in pennate diatoms which are generally heterothallic (they have two separated sexes).

 In Vanstechelman et al, 2013, the authors generated mating type-specific linkage maps for the benthic pennate diatom Seminavis robusta. The mating type phenotype was mapped as a monogenic trait, and the work represented the first evidence for a genetic sex determining mechanism in a diatom, although the boundaries of the specific region and the specific gene/s involved have not been found. The second paper is our own publication, Russo et al., 2018, in which we used a transcriptomics approach to identify mating type related genes in the pennate planktonic diatom Pseudo-nitzschia multistriata, and demonstrated that one of these genes, MRP3, acts as a sex determinant. MRP3 is able to specify the mating type plus (MT+) and unfortunately has no homology with known genes, therefore we do not know its specific function in the cell. There is no conservation between P. multistriata and S. robusta in terms of the genes contained in the mating type locus, and our personal belief is that mating type systems in diatoms have evolved multiple times. Moreover, the P. multistriata mating type related genes are not conserved in centric diatoms (Russo et al., 2018).

Nothing is known for centric diatoms which are generally homothallic, like S. marinoi.

Are the genes that are involved in pheromone production or sensing known, or might these be among the genes that are induced here?

Again, the only available studies for diatoms have been carried out in S. robusta and P. multistriata, in which some genes potentially involved in pheromone production and sensing have been identified (see Moeys et al., 2016 and Basu et al., 2017). The most important ones in these two species do not have a wide conservation and many lack any annotation.

It is very likely that some of the genes induced in S. marinoi are required for pheromone production or sensing, and more work is required to provide support. We highlighted the conserved induced genes between P. multistriata and S. marinoi (the SIG genes in Table 3) because these could be some of the candidates to follow. Pheromones are supposed to be species-specific to avoid breeding between cells of different species, however processes downstream of the response to the very first signals might be more conserved across species.

We added a comment in the discussion (lines 457-460).

Vanstechelman I, Sabbe K, Vyverman W, Vanormelingen P, Vuylsteke M (2013) Linkage Mapping Identifies the Sex Determining Region as a Single Locus in the Pennate Diatom Seminavis robusta. PLoS ONE 8:e60132 . doi: 10.1371/journal.pone.0060132

Moeys S, Frenkel J, Lembke C, Gillard JTF, Devos V, Berge KV den, Bouillon B, Huysman MJJ, Decker SD, Scharf J, Bones A, Brembu T, Winge P, Sabbe K, Vuylsteke M, Clement L, Veylder LD, Pohnert G, Vyverman W (2016) A sex-inducing pheromone triggers cell cycle arrest and mate attraction in the diatom Seminavis robusta. Scientific Reports 6:19252 . doi: 10.1038/srep19252

Basu S, Patil S, Mapleson D, Russo MT, Vitale L, Fevola C, Maumus F, Casotti R, Mock T, Caccamo M, Montresor M, Sanges R, Ferrante MI (2017) Finding a partner in the ocean: molecular and evolutionary bases of the response to sexual cues in a planktonic diatom. New Phytol 215:140–156 . doi: 10.1111/nph.14557

Russo MT, Vitale L, Entrambasaguas L, Anestis K, Fattorini N, Romano F, Minucci C, Luca PD, Biffali E, Vyverman W, Sanges R, Montresor M, Ferrante MI (2018) MRP3 is a sex determining gene in the diatom Pseudo-nitzschia multistriata. Nature Communications 9:5050 . doi: 10.1038/s41467-018-07496-0

Reviewer 2 Report

I have attached a document with the comments and suggestions for the reviewers- it is titled "Reviewer Report.pdf"

Author Response

REFEREE #2

Reviewer Report: Article genes-510672

Summary:

In the article entitled “Exploring molecular signs of sex in the marine diatom Skeletonema marinoi”, the authors compared the transcriptomes of sexually induced strains of the diatom with uninduced strains. The authors aimed to investigate the genes involved in sexual development in this species with an effort to potentially identify genes that could be used as markers to indicated sexual reproduction in metagenomic and metatranscriptomic datasets. The results include the identification of a number of genes related to sexual processes, such as meiosis as well as those involved in flagellar formation. Additionally, the authors define a set of unannotated genes as “sexually induced genes” due to their differential regulation during sexual development. They were further able to observe differential expression of a subset of these genes. They thus conclude that this study has expanded the set of genes that can be used as a marker for sexual reproduction, particularly for diatom speciess. I believe this manuscript fits well within the scope of Genes, particularly with regards to the study of gene expression, genomics and transcriptomics. The paper also has the potential to make a significant contribution to this specific field, particularly if the research is presented in a more understandable and coherent manner.

We thank the reviewer for his/her positive comments.

My overall impression of this manuscript is that although the science is mostly sound and that the analyses conducted are appropriate for transcriptomic data, a number of improvements could be made to make it more suitable for publication purposes.

Broad Comments:

1. Major concern 1- The transcriptomic comparisons:

• The authors present the transcriptomes of three different cell types, 1) Small cells with salinity stress (sexualized cells), 2) Large cells with salinity stress (non-sexualized cells) and 3) Large cells without salinity stress (control). It would also have been appropriate to include a fourth cell type- small cells without salinity stress, ie: small cells that are not sexually competent. The manuscript’s comparison between the sexualized and non-sexualized cells allows for genes involved in cell size to be detected as differentially expressed between the two cell types, which then may be incorrectly considered as sexually-induced genes.

We agree that the cell size is a confounding factor in the experiment and comparisons that we made, and that a comparison between large and small cells without any stress would have added another important layer of information (the genes that change in dependence of the cell size). Unfortunately we did not include this fourth condition. Some of the genes that distinguish small cells from large cells are certainly part of the DEGs in the Sex comparisons, however we believe that a great part of the DEGs are changing because of the major change occurring in the cell population (the switch from mitosis to meiosis in some cells and the expression of other genes related to signaling during the sexual phase). Indeed there is also an increase in the number of DEGs when more sexual stages have been produced (4807 at T1 and 8528, almost double, at T2 for SEX, while for the SALT comparison it is from 1955 to 2500). Moreover, in the manuscript we place most of the focus on genes which are logically linked to sexual reproduction (tables 1-3). In any case, we have added a comment in paragraph 4.3 in the discussion (lines 469-478) to indicate that we cannot exclude that some of the other general changes that we observed (such as those described in paragraph 3.5) could be due to the different gene expression profiles of large versus small cells, and that a dedicated experiment to explore this difference will be needed to refine our knowledge of the process.

• It is fairly difficult to keep track of the various comparisons made between the numerous datasets. For example, in text (line 140), it states that “four pairwise comparisons” were conducted, however, in table 1, there are five listed comparisons.

We have added Figure 2 NEW that illustrates the comparisons that were carried out. We have eliminated Table 1 and have presented the data on DEGs for the different comparisons in Figure 3 NEW. We removed the T1/T2 comparison originally present in Table 1 as it was not discussed anywhere in the text.

Perhaps Figure 1 could be improved and added to by including the various sample types that were collected at T1 and T2, and what comparisons were done. Alternatively, an entirely new figure could be generated.

We agree that our description of the experiment and of the comparisons was not very clear. We have created a new Figure 2, which we hope will help in improving clarity.

• The abbreviations given to each comparison are not intuitive and perhaps are not entirely necessary. Furthermore, using the abbreviations “SST1” and “SST2” to refer to the salinity stress timepoints 1 and 2 is confusing given that “SST” is used as the abbreviation for “sexualization size threshold” earlier in the manuscript (lines 45). Perhaps use something like “SS_T1”.

Yes SST is confusing, we have changed to SALT_T1 and SALT_T2.

2. Major concern 2- Discussion involving the flagella genes:

• This concern is also linked to the comment 9 below regarding the time points selected. It is originally made clear (line 189) that swimming, assumingly flaggelate, gametes were only observed more than 62 hours after transferal to the salinity stress media. However, later in the manuscript (line 260), the authors state that the differential expression of flagella genes is consistent with the finding of flagellate gametes at T2. T2 is significantly earlier than the 62 hours after which flagellate gametes are reported. This statement is made again in the discussion (line 326). This needs urgent clarification.

We are sorry if there was an apparent inconsistency. In the experiment run the week before the transcriptomic sampling, described in old Figure 1 and now in new Figure 2E, we recorded the timing of appearance of the sexual stages only in light microscopy that does not allow to visualize the nuclei. In previous experiments, where we used the nuclear stain SYBR green, we observed cells containing four nuclei, i.e. male gametangia, at the same time point in which we observed the bent cells interpreted as female gametangia, on day 2 after the application of the salinity cue. Thus we reasoned that early on day 3 (T2), some cells were already engaged in male meiosis and therefore had turned on the genetic program needed to build flagella. We slightly modified the sentence at lines 301-302.

• In both the abstract (lines 21 – 22) and the discussion (lines 313 - 317), the fact that genes involved in flagellar synthesis were not found to be expressed in S. marinoi despite having been found in L. danicus was specifically referenced. Why is this particular result focused on so much? Especially considering genes involved in flagellar synthesis in E. huxleii were found to be differentially expressed in S. marinoi.

We were surprised because the IFT proteins are conserved in eukaryotes and the absence of many of them was unexpected. IFT proteins were expressed in both L. danicus and in E. huxleyi, but not in S. marinoi. Some of these genes were also not recovered in the genome of S. marinoi.  To us this is an indication that the way flagella are built and maintained in the two diatom species could be different. We have have restructured the flagellar genes paragraph (4.2) to make it more effective.

3. Major concern 3- Section 2.6:

• This section is very difficult to follow and not written in a logical manner. I think the entire section needs to be rewritten in an effort to better this manuscript. It appears that slightly different steps were taken for genes from L. danicus vs genes from P. multistriata, this may well be appropriate, but it could be better explained. The last paragraph of this section also needs some clarity as the steps of the method are difficult to follow.

We have re-written part of this section and have added a new supplementary figure (S2) that explains the analysis for the P. multistriata dataset step by step.

4. There are too many tables and not enough figures in this manuscript. While transcriptomic data often calls for the use of tables and is appropriate in certain cases, some of the data presented here could be better visualized in figure form. One such example is Table 1. Perhaps including a figure that compares sexualized and nonsexualized cells (could be indicated by small and large cells, respectively) at the two time points with the number of up- and down-regulated genes indicated on the figure. Additionally, the GO term enrichment analyses performed provide perfect data for figure-based visualization.

We thank the reviewer for the suggestions. We modified the old figures and have new figures in the current version of the manuscript, specifically three new figures in the text and two new figures in the supplementary materials (Fig. 1 NEW: schematic drawing of the life cycle of centric diatoms; Fig. 2 NEW: the experimental set up; Fig. 3 NEW: the DEG genes in the different comparisons; Fig. S2 NEW: the workflow for the comparison between DEG genes in P. multistriata  and S. marinoi; Fig. S3 NEW: GO terms for SEX_T1, SEX_T2, SALT_T1 and SALT_T2). We hope that they improve clarity and aesthetics of the paper.

5. The introduction is not comprehensive enough if the authors wish to appeal to the broader audience of this journal. A more in depth description of sexual reproduction in diatoms would be helpful, perhaps including a figure to show the various stages of sex in these species. For example, the unique manner of the progressive size reduction and “the rigid fustule” the authors refer to are not as clear as they could be. Additionally, a short description of the difference between pennate and central diatoms would give good context for some of the comparisons to other species which are made later in the manuscript.

We modified the text in the introduction and added a new figure (Figure 1 NEW) with a schematic illustration of the life cycle of centric diatoms.

6. The paragraphs beginning on lines 48 and 57 could be combined and rewritten to form a more logical “story”. Alternatively, two independent paragraphs can be written with this information as the two paragraphs have a mixture of information regarding chemical sensing and genes involved in sexual reproduction.

This section has been rewritten (lines 73-84).

7. The final paragraph of the introduction, where the aim of the study is discussed, it packed with unnecessary detail that takes away from the main message. This paragraphs should be written more concisely. The extra information could either be moved to somewhere in the introduction, methods and material or discussion, where appropriate.

This section has been slightly modified.

8. In two cases, the databases used were accessed a number of years ago. In line 133, for example, the UniRef90 database was accessed mid-2016 while in line 166 the NCBI NR database was accessed early in 2017. It would be appropriate to relook at these databases and determine whether new data has been published in these databases that may be of use.

We run the analyses again using the versions of the databases downloaded in May 2019, as now indicated in the Methods. For the UniRef annotation of the 39,660 contigs, 705 of them have names which differ from before. However, some of these names appear to be either alternative names for the same gene, or minor spelling changes (such as the addition of a hyphen). We introduced small changes to annotations in the Tables for genes for which a difference had been found. We provide the complete Annotation file as supplementary material.

In the case of the second analysis, using the new version provided the same result.

9. Under section 2.1 Experimental set up, the authors state they do a pilot test to determine the correct time points for the transcriptomic study, by looking at timepoints 20, 30, 40 and 62 hours post transfer (line 92). Shortly thereafter (line 100), the authors state that Timepoint 2 will be at 52 hours- why? There is no apparent justification for the choice of this time point.

See our reply to ‘Major concern 2’.

10. In lines 205 – 208, the authors state that they expected to observe the induction of certain genes that were detected in a similar study in P. multistriata. Why are the other comparative species not mentioned here?

There are not many investigations dealing with sexual reproduction in diatoms. The other major paper reporting a transcriptomic study during sexual reproduction has been published by Moeys et al. in 2016. In this paper, the effect of the Seminavis robusta MT+ pheromone (SIP+) on MT- cells is explored by collecting MT- cells very early in the process, 30 minutes, 1 hour and 3 hours after the addition of the purified SIP+. This experimental set up is quite different from the one used in our manuscript. The set up used for P. multistriata was also different, but in this case the response was evaluated for both mating types and slightly later (6 hours).

In addition, we have a much deeper knowledge of the P. multistriata dataset since we have generated the data and currently work on the system, while for Moeys et al. not all the results are discussed, moreover S. robusta raw data have been deposited in a public database but the assembled S. robusta transcriptome and the complete table of DEGs have not been made available.

11. Late in the manuscript, the authors refer to the genome sequence of this species. If such data exists, why was a de novo transcriptome assembled? Why did the authors not simply align the RNA reads to the genome in order to determine differential expression etc?

The work presented in the current manuscript was initiated before the annotation of the reference genome sequence had been done. A de novo assembly approach was therefore preferred for the differential expression analysis.

12. The results section from lines 224 – 233 is insufficient. The authors refer to “significantly enriched GO terms” but do not state what these are. A figure would be more descriptive and informative. Furthermore, GO terms can be broken down into various categories- cellular component, molecular function and biological process. Indicating which of these categories are enriched would also provide better information and more understandable results.

We added new supplementary Figure S3 with the GO terms broken into categories and added a few more details in the text (lines 267-272).

13. In table 2, the query IDs are from Arabidopsis and Saccharomyces- why is this the case? Are there not more appropriate, closer relatives to use?

These IDs were the ones chosen by Patil et al. (2015) and were easily retrieved. Arabidopsis was originally chosen by Patil because it is a model system in which meiosis has been extensively studied and characterized, and has been used as a reference in similar works. Retrospectively we could have used diatom sequences from the same paper, but we have retrieved all the expected sequences plus BRCA1, which is an independent confirmation of the toolkit composition.

14. In lines 248 – 253, the absence of the GEX1 and HAP2 genes are indicated. This is briefly brought up again in the discussion, but I think it needs to be discussed more, given the importance of these genes to other eukaryotes and the suggestion that these genes be used as the markers for eukaryotic sex.

We added a comment in the discussion (lines 487-490).

15. The paragraphs starting on lines 365 and 373 seem somewhat irrelevant to the overall discussion of this manuscript. I’d suggest either integrating this information with the data collected here or excluding it entirely.

We modified the paragraphs shortening them and linking the information with the S. marinoi data.

16. An interesting finding in this manuscript is the general down-regulation seen in genes associated with the chloroplast during sexual reproduction. The paragraph beginning on line 412 very briefly discusses this. I suggest going into more detail discussing this result and providing some kind of plausible explanation.

Yes we also believe that this is an interesting signal, we see a similar downregulation of photosynthesis related genes in P. multistriata during the late phases of sexual reproduction (unpublished new data), however in P. multistriata we have a marked growth arrest as a consequence of pheromone signaling (not observed in S. marinoi, see lines 463-464), therefore downregulation of photosynthesis in P. multistriata reproducing cultures could be a consequence of cells arresting growth and reducing metabolic rates. Alternatively, the downregulation of photosynthesis might be directly linked to sexual reproduction as it seems to happen in S. marinoi as well. Since we do not have a definitive understanding of these observations, it is currently hard to speculate. We highlight the findings in the current paper for future reference (lines 464-468) but would like to be cautious on providing explanations. We are planning measurements of photosynthetic parameters to confirm the transcriptomics signal.

Specific Comments:

1. Consistency:

• In section 2.2 RNA extraction and RNA-seq, the single-end and pair-end libraries are abbreviated to SE and PE (lines 107 and 111, respectively). This abbreviation, however, is not used consistently throughout the rest of the document. This should be fixed (eg: lines 120, 123, and 129).

Done.

• When referring to E-values, both capital and lowercase E’s are used to denote the scientific numbering format (eg: Line 158: E-value of 1E-02 vs Line 240: (Evalue < 1e-5). Please ensure this is consistent throughout the document.

Done.

• It is general notation to write gene names in italics (eg: Msh4) and protein names in capital letters and standard font (eg: MSH4). This should be standardized throughout the manuscript.

Correct, we verified that the names are respecting this notion. 

• Both “up-regulated, down-regulated” and “upregulated, downregulated” are used. Please standardize throughout the manuscript. • Numbers in the excess of 1000 are referred to in the manuscript either as “1000” or “1,000”. Please standardize throughout the manuscript.

Done.

2. Line 36: “a box and its lid” is an unnecessary comparison.

Removed.

3. Lines 40 – 42: The definition of homothallic mating is not particularly descriptive of the system, especially in comparison to heterothallic mating. This definition should include a clear indication that homothallic mating occurs in the absence of mating partners. Alternatively, given that the type of mating occurring is irrelevant to the overall aims and results of this paper- these definitions could be removed.

We do not think that this information is irrelevant. The heterothallic mating system is what is called ‘dioecious’ in plants and the homothallic mating system corresponds to ‘monoecious’. We have added in parenthesis these two more familiar terms. The correct ones for diatoms are homothallic and heterothallic (see Kaczmarska, I., Poulíčková, A., Sato, S., Edlund, M. B., Idei, M., Watanabe, T. & Mann, D. G. 2013. Proposals for a terminology for diatom sexual reproduction, auxospores and resting stages. Diatom Res.:1-32).

4. Line 86 and other: The abbreviation for sexualization size threshold (SST) is indicated in line 45. In other places where this comes up, please use the abbreviation instead of the entire term.

Done.

5. Lines 107 – 110 and 111 – 112: The detail provided for the preparation of the SE libraries was perfect. However, the same level of detail is not provided for the preparation of the PE library. This should be supplemented. Furthermore, it is clear why the SE libraries were prepared, but why were the PE libraries made as well?

Perhaps a sentence starting “In order to **, we also sequenced a PE library…”.

The text has been updated with more details regarding the library preparation, as well as a statement that this dataset was generated to facilitate the de novo transcriptome assembly (lines 156-158).

6. Line 112: Why was a different S. marinoi strain used for the PE library? The origin of this isolate should also be indicated in the experimental set up.

The strain ST54 is one of the reference strains we often use for analysis of S. marinoi (see e.g. Töpel et al. 2018, Journal of Genomics, vol. 6 pages 113-116). A statement of its origin has been added to the Experimental setup section (lines 107-109).

7. Line 114: What is a “summary control”?

This sentence has been rephrased to indicate that it is “quality control” we mean (line 160).

8. Line 114 and others: There are many instances in the manuscript where particular software packages are used but not referenced. Please add references to these where appropriate. (eg: FastQC software is used and should be referenced as: “Andrews, Simon. FastQC: a quality control tool for high throughput sequence data. (2010)”.

We have now included the appropriate references for the software FastQC, Bowtie2 and RSEM.

9. Line 127 – 130: A single sentence spans these lines and is confusing to understand. Please break it up into two or more independent sentences for better understanding.

This part has now been split on two sentences.

10. Line 130: Please provide an explanation for “positive contig impact score” and explain in more detail why these contigs were also added to the final analyses.

The positive contig impact score is reported by the program Transrate, and this is now stated in the text (lines 176-177). We included transcripts in the analysis because of the positive score reported by Transrate, which makes us believe they are true transcripts.

11. Line 128: Instead of providing the URL for the genome sequence used, instead provide the two references that relevant for this genome (Bowler et al. 2008 & Armbrust et al. 2004).

The Phaeodactylum genome was not used here. We added the reference but left the URL since it refers to a specific version of the gene models (v3).

12. Line 136: The authors refer to the biological replicate. It is not made clear beforehand what represses a biological control, even in Figure S1.

This detail has been added in the Methods (lines 129-130) and in Figure 2 NEW.

13. Line 139: “… at least above 1 CPM...”. What is CPM in this context and how is it calculated?

CPM stands for Counts Per Million and this has been clarified in the manuscript.

14. Line 140: “… was used to identify differentially expressed genes for each…”- This is the first reference to differentially expressed genes and the DEG abbreviation should be used here so that it can be used elsewhere (eg: line 229).

Done.

15. Lines 141 and 142: “…T1 and T2: i) large cells…”- What does the i) refer to? There is no follow up ii) so it seems unnecessary.

Removed.

16. Lines 143, 144 and others: The logFC changes are referred to as “-1<logFC<1” throughout the manuscript. This is not correct. It should be indicated as “-1>logFC>1” as significant differences are usually considered if the fold change is BELOW -1 or ABOVE 1.

Corrected.

17. Lines 154 and 155: “The list of meiotic-related genes was taken from [7] and the one for flagella [9] and [10].”- Please indicate from which species these genes were taken, it will provide better context that simply providing the references.

Species added.

18. Line 165: “…S. marinoi were blasted (BLASTx)…”- One cannot use “BLAST” as a verb as in this sentence, as in “blasted”. Please rephrase so that this is not the case.

The whole paragraph has been rewritten.

19. Line 176: The authors simply state “Bicosoecid” despite the fact the sentence is referring to a particular species. Which species is specifically being used?

It is actually an unidentified Bicosoecid. In the MMETSP database (https://www.imicrobe.us/#/samples/1729) it is indicated as Bicosoecid sp, strain ms1, which is what we add in the text (line 222).

20. Line 188: The authors refer to the gametangia as “elongated cells with a bent shape”. Instead of this description, could the authors state “gametangia, which represent an early sexual tissue” or something similar?

We added “which will produce gametes”.

21. Line 189: The authors refer to the auxospores as “round structures still attached to the gametangial cells”. Instead of this description, could the authors state “auxospores, which are the sexual cells that result after cell fusion” or something similar?

We wrote “auxospores, i.e. round structures which result from gametes fusion, still attached to the ‘female’ gametangial cells,…”. In the new version there is a definition in the legend of Figure 1 NEW too.

22. Line 202 onwards: The first paragraph of this section of results has information better suited to the methods and materials section. It should be rewritten with only the results.

We have modified the paragraph and have included details about the choice of the time points.

23. Lines 207, 263, 269, 280 and others: There are many instances in the manuscript where species names have not been italicized.

There must have been a formatting issue in the very last version, we should have all the names in italics now.

24. Line 209: “We sequenced 14 samples…”- Although these 14 samples are explained in Figure S1, it should be made clearer in the main text. Furthermore, why are there only 2 replicates for the Large cells (both salt and non-salt), but 3 replicates for the Small cells? Transcriptomics projects that are not followed up with a confirmation experiment (like RT-qPCR) usually have a minimum of 3 repeats per condition.

We added the information in the Methods (lines 129-130) and in Figure 2 NEW. Unfortunately technical issues prevented us from sequencing three replicates for two of the conditions. We agree that the best practice would be to use an independent method to validate transcriptomics data (although this was mostly strictly required for microarrays and now less used for RNA-seq where the large amount of data makes statistics stronger), indeed we have generally done this in previous projects, however we could not perform any qPCR in this case. In the current manuscript we discuss mainly gene sets which have to do with sexual reproduction and place less emphasis on general changes except very strong signals, therefore we believe that the main results and conclusions presented are still valid.

25. Line 237 and 238: The authors refer specifically to genes BRCA1, SMC2 and SMC4. What are these genes, what kind of proteins do they encode and what role do they have in sexual reproduction?

We added information in the text (lines 277 and 279-281).

26. Lines 238 and 435: The authors refer to genes being “overexpressed”. Do they mean “up-regulated”? These two terms mean slightly different things.

We changed “overexpressed” to “upregulated”.

27. Table 2: Instead of using (1) and (2) to link to the footnotes, using a superscript a and b would be more readable and less confusing.

Change made.

28. Table 4: Why are some of the LogFC changes in bold text? Why do some of the contigs not have annotations (if they are unknown proteins, indicate as such)? What does -::- mean under the P. multistriata annotation mean? Indicating the genes that show the same direction of regulation by “^” is not very clear and these genes are difficult to identify. Perhaps indicating their LogFC in bold or italics, or with a superscript would be clearer.

LogFC in bold were the ones above 1 or below -1, in table we reported also values above 0.7 or below -0.7 for one of the two conditions to show that the variation is in the same direction. We removed the formatting. The – and -::- characters were those found in the original annotation tables. When the software used to find homologies fails to find any similar sequence it does not assign an annotation, we left the – character in both columns because it reflects absence of annotation. We replaced “^” with “(a)”, with the same style chosen for Table 1.   

29. Lines 261 – 269: The authors discuss the retrieval of genes related to flagella synthesis and state that eight of the L. danicus genes are found in the S. marinoi transcriptome and a further two in the genome, bringing the total number of homologous genes to 10. This suggests that the first eight were not found in the genome- is this the case? If so, how?

No, those eight are also in the genome, as shown in Supplementary Table S3. We meant that of the 14 missing in the transcriptome, only two were found in the genome, indicating that they exist in the species but were not transcribed at all in our samples. If we had found all of the missing ones in the genome, we could have hypothesized that the same set was shared between S. marinoi and L. danicus but that for some reason we had not seen expression in our samples. The absence in the genome is an important point as it supports the idea that the set of genes needed by S. marinoi to build flagella might be different from L. danicus.

We modified the text (lines 309-311) to avoid confusion.

30. Line 272: The comparison of sex-related genes from T. weissflogii is brought up for the first time here- perhaps this should be discussed earlier when the other comparative species are brought to attention.

We included a reference to T. weissflogii at the end of the introduction (lines 94-95).

31. Line 284: The identification of the SIG1, 2 and 3 genes is never brought up in the methods and materials section. How were these genes discovered in S. marinoi and what is their relevance?

We have added details on how the SIG genes were identified in the Methods (lines 209-211), and more information on their putative function in the discussion (lines 450-452).

32. Lines 290 – 299: These lines represent a single sentence which is difficult to follow and understand. Please rewrite into concise, independent sentences.

Yes, we broke the sentence in three parts (lines 332-343).

33. Line 300: “…can be found as highly regulated…”- What does highly regulated mean? Up-regulated? Significantly differentially expressed?

We changed to “strongly upregulated”.

34. Line 310: the authors state that their analysis was able to identify genes associated with the switch to meiosis. This is not strictly true. Although meiosis is an important factor in sexual reproduction, many other pathways and processes are taking place during this time. A more appropriate remark would be “… associated with the switch from the vegetative lifestyle to sexual development”.

We have changed into: “associated with switch from the vegetative lifestyle to sexual reproduction”.

35. Lines 327 – 328: Delete “ie genes for which functions outside meiosis are not known”- this is implied by the description “specific for meiosis”.

We changed the text to: “exclusively required, SPO11-2…”.

36. Lines 354 – 356: The authors need to provide a reference for this sentence.

We added a reference to a book where this information can be found.

37. Line 365: The authors have been consistent in describing the type of species, such as the “pennate P. multistriata”. This is helpful and should also be done for T. pseudonana and other species as it provides context for each comparison.

We added that it is a centric diatom and did the same elsewhere when appropriate.

38. Lines 385 and 396: The authors refer to the set of flagella genes as an “asset”. Do they perhaps mean “set”?

Yes it is set, it has been corrected.

39. Line 399: the abbreviation MMESTP comes up in this sentence. This abbreviation needs clarification.

That was a spelling mistake, it is MMETSP, mentioned in the Methods where we added the meaning.

40. Line 407: the term “mastigoneme” is used. This word would only be understood by a very particular audience and should either be excluded or, if relevant, explained.

We addede a sentence to explain what mastigonemes are (lines 453-354).

41. Grammatical/typographical errors:

• Line 15: “increase” should be “increased”

We mean that it contributes to increase the genetic diversity.

• Line 45: “cohorts” should be “cohort”

In our opinion, the plural has to be used here since we refer to the progeny of cells that derive (by vegetative division) from the initial cells.

• Line 46: “deriving” should be “derived”

OK.

• Line 50: The sentence should read “Chemical cues also play a role…”

Changed.

• Line 53: “to identify” should be “the identification of”

The relative sentence has been deleted in the new version.

• Line 58: “case of” should be “case for”

OK.

• Line 92: “transfer at higher” should be “transfer to media of a higher”

OK.

• Line 97: “adjusted at” should be “adjusted to”

OK.

• Line 102: It should be mentioned that it is RNA that is being extracted

Added.

• Line 161 & 177: “blast” should be “BLAST”

Corrected.

• Line 178: “an aligned” should be “and aligned”

Corrected.

• Line 187: “below SST transferred” should be “below the SST and transferred”

We think that “below SST transferred” works as well in this sentence.

• Line 225: “a few” should be “few”

OK.

• Line 256 and 258: The species name E. huxleyi is spelt incorrectly.

Corrected.

• Line 302: “supposedly” should be “putatively”

Change made.

• Line 354: “its absence is some” should be “its absence in some”

Corrected.

• Line 302: “Pennate diatoms…” should be “While pennate diatoms…” (to indicate comparison to centric diatoms)

Change made.

• Line 410: “will allow to reveal” should be “will reveal”

Change made.

• Line 411: “worth” should be “worthy”

Corrected.

• Line 440: There is an extra “B” before the supplementary materials are listed.

Corrected.

We truly thank the reviewer for the careful assessment of the manuscript.

Reviewer 3 Report

The overall objective of the paper by Ferrante et al. entitled “Exploring molecular signs of sex in the marine diatom Skeletonema marinoi” was to gain a better understanding of the sexual processes and regulation in diatom S. marinoi. Transcriptomic approach was used to identify genes related to the sexual phase, in particular considering genes involved in meiosis and in flagella formation.

The paper is interesting and well written. Title and abstract are appropriate for the content of the text. Introduction is clear and provides sufficient background about research topic. The right research methods have been selected. The results are presented in a proper way and they are discussed appropriately.

I consider that the article can be accepted after to revise some minor details:

-Results section: all species names in italics.

-Page 8 Table 2. Change the title of the last column with SST2 logFC.

-Authors performed a differential expressed genes analysis of large cells grown in standard condition versus large cells grown in higher salinity, however the detailed analysis of genes regulated by salinity stress alone has been postponed to a future study. These results on salinity stress is very promising, but too preliminary, not properly discussed and not essential for the final goal of the study. I suggest to remove this comparison (also in Table 1 and in Veen diagram) and going into more detail in a future study.

Author Response

REFEREE #3

Comments and Suggestions for Authors

The overall objective of the paper by Ferrante et al. entitled “Exploring molecular signs of sex in the marine diatom Skeletonema marinoi” was to gain a better understanding of the sexual processes and regulation in diatom S. marinoi. Transcriptomic approach was used to identify genes related to the sexual phase, in particular considering genes involved in meiosis and in flagella formation.

The paper is interesting and well written. Title and abstract are appropriate for the content of the text. Introduction is clear and provides sufficient background about research topic. The right research methods have been selected. The results are presented in a proper way and they are discussed appropriately.

We thank the reviewer for the positive evaluation of our manuscript.

I consider that the article can be accepted after to revise some minor details:

-Results section: all species names in italics.

We have placed all species names in Italics, there must have been a formatting problem in the previous version.

-Page 8 Table 2. Change the title of the last column with SST2 logFC.

We have now changed SS to SALT, and we corrected the mistake in the table.

-Authors performed a differential expressed genes analysis of large cells grown in standard condition versus large cells grown in higher salinity, however the detailed analysis of genes regulated by salinity stress alone has been postponed to a future study. These results on salinity stress is very promising, but too preliminary, not properly discussed and not essential for the final goal of the study. I suggest to remove this comparison (also in Table 1 and in Veen diagram) and going into more detail in a future study.

We agree that a more in depth study is needed for the investigation of the effects of the salinity shift, we believe however that we cannot completely exclude the data from the present manuscript since we need the salinity data to demonstrate that the responses that we highlight are not a general reaction to the environmental cue but are rather linked to the activation of the sexual reproduction program (see for instance columns with FC in new Tables 1 and 2).

Reviewer 4 Report

This study explores genes that are expressed in a diatom during growth conditions that support formation of gametes and sexual reproduction.   Genes associated with the meiotic toolkit were found to be expressed, whereas flagellar genes were not found to be expressed.  The first result is consistent with sexual reproduction, whereas the later result is somewhat unexpected given results from previous studies of other species.  This study is fine as far as it goes and may be what is possible in this system/organism.  What would make this of greater impact is if the introduction and discussion covered advances in what is known about other stremenopiles such as oomycetes, or other organisms such as fungi and algae.  The authors could consider for example citing a recent chapter by Goodenough and Heitman that covers an evolutionary perspective on sexual reproduction:

Origins of eukaryotic sexual reproduction.

Goodenough U, Heitman J.

Cold Spring Harb Perspect Biol. 2014 Mar 1;6(3). pii: a016154. doi: 10.1101/cshperspect.a016154. Review.

Specific questions.

Has a mating type locus or sex chromosome been identified in this or any other diatom?

Are the genes that are involved in pheromone production or sensing known, or might these be among the genes that are induced here?

Author Response

(The authors gave the same response as above.)

Round 2

Reviewer 1 Report

Thanks for this improved version of the manuscript. I consider it suitable for publication. 

I am still a bit confused by the experimental design. I think a control condition with small cells in low salt condition is the only way to control for a cell size effect on gene expression. Maybe you could add a comment on this aspect. 

Reviewer 2 Report

To the authors:

Thank you for your detailed reply to my comments and suggestions. I believe the manuscript to be much improved, easier to read and I enjoyed looking through it again. 

The additional figures you provided also made the experimental set up and results easier to understand. 

Reviewer 3 Report

The authors have satisfactorily improved the manuscript and I consider that it is now suitable for pubblication.

Reviewer 4 Report

The authors have adequately addressed the points raised by review and the manuscript is fine to accept and publish now from my perspective.